# Epigenetic Regulation of Autophagy in Cardiovascular Pathobiology

**DOI:** 10.3390/ijms22126544

**Published:** 2021-06-18

**Authors:** Shuhan Bu, Krishna K. Singh

**Affiliations:** Department of Medical Biophysics, Schulich School of Medicine and Dentistry, University of Western Ontario, London, ON N6A 5C1, Canada; sbu4@uwo.ca

**Keywords:** epigenetics, autophagy, atherosclerosis, cardiovascular diseases

## Abstract

Cardiovascular diseases (CVDs) are the number one cause of debilitation and mortality worldwide, with a need for cost-effective therapeutics. Autophagy is a highly conserved catabolic recycling pathway triggered by various intra- or extracellular stimuli to play an essential role in development and pathologies, including CVDs. Accordingly, there is great interest in identifying mechanisms that govern autophagic regulation. Autophagic regulation is very complex and multifactorial that includes epigenetic pathways, such as histone modifications to regulate autophagy-related gene expression, decapping-associated mRNA degradation, microRNAs, and long non-coding RNAs; pathways are also known to play roles in CVDs. Molecular understanding of epigenetic-based pathways involved in autophagy and CVDs not only will enhance the understanding of CVDs, but may also provide novel therapeutic targets and biomarkers for CVDs.

## 1. Introduction

Macroautophagy or autophagy is a highly conserved pathway to maintain cellular homeostasis via cellular turnover of unnecessary proteins and organelles that are better used to sustain the cell’s survival [1]. Autophagy is a crucial regulator of various biological processes and is closely associated with the pathobiology of cardiovascular diseases (CVDs). Autophagy is tightly regulated at various levels and epigenetic regulation by a class of non-coding RNAs, such as by microRNAs (miRNAs) and long non-coding RNAs (lncRNAs). This review will discuss the current understanding of the epigenetic regulation of autophagy, particularly in relation to the pathobiology of cardiovascular systems. In the end, we will discuss the application of epigenetic modulation of autophagy as potential therapeutic targets for the treatment of CVDs.

## 2. Molecular Mechanisms of Autophagy

There are several reviews on this topic [2,3,4], however, briefly during the autophagic process, the cellular components are internalized into a double membraned autophagosomes, which ultimately fuses with lysosomes for degradation into materials, such as amino acids [5]. Autophagy is usually activated by nutrient deprivation [6], and autophagy activation is a three-step process that involves induction of autophagy, nucleation of the autophagosome, and elongation of the autophagosome. To successfully induce autophagy, inhibition of the mammalian target of rapamycin (mTOR) is required, activating unc-51-like kinase 1 (ULK1). ULK1 then phosphorylates other proteins, including autophagy-related gene 13 (ATG13), RB1-inducible coiled-coil1 (RB1CC1), to induce autophagy. Under nutrient-rich conditions, mTOR inhibits the activity of ULK1 and thereby inhibits autophagy [7]. Adenosine monophosphate-activated protein kinase (AMPK) is another protein involved in autophagy induction [8]. Under glucose deprivation, AMPK phosphorylates the tuberous sclerosis complex (TSC1/2), inactivates RHEB, a GTPase activating protein, and inhibits mTOR leading to autophagy stimulation [9]. The next stage is membrane nucleation which leads to the assembly of the phagophore. The class III phosphatidykinositol-3-kinase (Ptdlns3K) complex contains Ptdlns3K vacuolar protein sorting 34 (VPS34) and generates phosphatidykinositol-3 phosphate. Many autophagy-related genes (ATGs) have been identified to be essential for the formation of the autophagosomes. For example, ATG7 is a central regulator of endothelial autophagy as knocking down ATG7 leads to impaired autophagic influx and inhibition of autophagy [10]. The formation and elongation of the autophagosomes involve two ubiquitin-like conjugation systems. The first conjugation system is the ATG5-ATG12 conjugation system formed by the protein ATG5 conjugating to ATG12 with the help of ubiquitin-like enzymes ATG7 and ATG10, which promotes the elongation of autophagosomal membrane. The second conjugation system involves microtubule-associated protein 1 light chain 3 (LC3), which is cleaved by the protease ATG4B and transformed to its mature form LC3-I, conjugating with phosphatidylethanolamine (PE) to form the lipidated version LC3-II [5]. The ratio of LC3-II/LC3I has, thus, become a common method to measure autophagic activity in cells [11]. Next, p62 associates with the ubiquinated cargos, and the complex interacts with LC3 on the autophagosome. After the completion of autophagosome formation, ATGs get recycled in the cytoplasm for reuse. Autophagosomes fuse with the lysosome to degrade the cargo by the lysosomal enzymes (Figure 1) [7].

## 3. Autophagy in Cardiovascular Pathobiology

Cardiovascular disease (CVD) is the leading cause of mortality that accounts for ~30% of all death worldwide [12]. The most important players in the pathogenesis of CVDs are vascular endothelial cells (ECs), which form the inner most layer of every blood vessel and function to maintain normal homeostasis in cardiovascular systems [13]. Endothelial cell apoptosis and dysfunction can lead to various CVDs, such as atherosclerosis and HF [14], and autophagy is shown to regulate both endothelial cell survival and function at baseline and under stress [13]. For example, von Willebrand factor (vWF) is secreted by ECs upon injury and stored in Weibel-Palade bodies (WPBs) to be later secreted into blood circulation, when required. Platelets recognize and bind to vWF, initiating platelet adhesion and the hemostasis cascade. It has been found that genetic or pharmacological inhibition of autophagy impairs the ligand-stimulated release of vWF, which disrupts vascular homeostasis [15]. Autophagy can also protect ECs by altering their energy metabolism [16]. Research has shown that increased autophagic influx could lead to increased lipid catabolism and clearance. Curcumin, an antioxidant, was found to induce autophagy in ECs and increase cell viability by increasing lipid catabolism [16]. In the context of autophagy-dependent lipid degradation, lipid droplets undergo lipolysis in lysosomes [17]. We have previously reported that both oxidized (OxLDL) and native low density lipoprotein (LDL) stimulates the formation of autophagosomes in primary ECs, and excess LDL particles are eliminated by autophagic vesicles [18]. The loss of autophagy by silencing ATG7 in both acute and chronic in vivo model resulted in accumulation of OxLDL in ECs, suggesting an essential role of autophagy in regulating excess lipids in the vessel wall [18]. Accumulation of OxLDL is one of the most common factors that promotes the build-up of atherosclerotic plaque [19]. These results suggest a beneficial or protective role of autophagy in CVD. At the same time, in the endothelial cell-specific autophagy-deficient mice, we observed reduced cardiac fatty acid storage, as well as reliance on fatty acid oxidation as a cardiac fuel source [20]. Autophagic inhibition in ECs is shown to induce TGF-β signaling and endothelial-to-mesenchymal transition, which is associated with loss of endothelial function and development of fibrotic diseases [10].

Autophagy can protect cells from oxidative stress, which plays an important role in the pathogenesis of CVDs. Previous studies revealed a link between mitochondrial reactive oxygen species (ROS) production and autophagic influx, or more specifically, mitophagic flux [21]. Mitophagy is a type of autophagy that is involved in removing damaged mitochondria from the cells to prevent the elevation of ROS in vascular ECs [22]. Increased ROS production serves as a stress signal to cells, resulting in increased autophagy by activating AMPK [23]. Defects in mitophagy and dysregulation in the redox state could lead to uncontrolled oxidative stress [24] and CVDs. The induction of autophagy is cytoprotective in smooth muscle cells in the presence of lipid peroxide [25]. Lipid peroxide triggers the ER stress response and activates autophagy via MAPK/JNK-dependent signaling [26]. Angiotensin II (Ang-II) is a well-known factor that contributes to the development of various CVDs [27]. It not only exerts vasoconstrictive effects on cardiomyocytes, which has been shown to induce myocardial ischemia, but it also affects the oxygen metabolism leading to augmented lipid peroxidation and oxidative stress causing coronary heart disease [27]. Shan et al., found that in Ang-II treated cultured ECs, inhibition of autophagy resulted in increased senescence and vascular endothelial cell damage [28]. Accordingly, we also observed an increased Ang-II-induced abdominal aortic aneurysm smooth muscle cell-specific autophagy in ATG7 deficient mice [29].

Shear stress in the vessel wall, which also plays an important role in the pathogenesis of CVD, is shown to be associated with autophagy [30]. Shear stress-induced autophagy has been found to increase cell viability, reduce oxidative stress, and increase nitric oxide synthesis by upregulating endothelial nitric oxide synthase (eNOS) [31], and accordingly, a direct correlation has been established between autophagy and eNOS production [32,33,34]. ENOS is an essential regulator of endothelial function and endothelium-mediated vasoreactivity [35]. Thus, autophagy inhibition will lead to reduced eNOS production, which will cause endothelial dysfunction and impair vascular homeostasis. In autophagy-deficient cells, shear stress cannot induce eNOS production, which results in increased ROS production and cellular inflammation by stimulating the production of cytokines; a potential cause for CVDs, such as diabetes and hypertension [35].

Obesity caused by a high fat diet (HFD) presents a significant burden to the heart and is another common cause of CVDs [36]. Autophagy also directly correlates with the degree of obesity and visceral fat distribution [37]. Akt2, which is an activator of mTOR, is activated under HFD in the heart, and knocking out Akt2 prevented cardiac hypertrophy and preserved cardiac function by promoting maturation of autophagosomes in the heart [38]. Furthermore, caloric restriction, which induces autophagy, has been shown to be a promising approach to treat CVDs by improving vascular function and slowing down the process of vascular aging via upregulating autophagy [39]. Autophagy is also involved in other cardiovascular stress conditions, such as ischemia/reperfusion (I/R). Using an intraoperative model developed by Sellke and colleagues, we found that there were 11/84 autophagy-related genes being upregulated in human ischemia and reperfusion injury and 3/84 being downregulated, indicating a role of autophagy in CVDs [40]. Overall, autophagy appears to be essential for maintaining cardiovascular homeostasis, and imbalance in autophagy is detrimental towards CVDs, whereas a controlled modulation of autophagy appears to provide a therapeutic intervention to treat CVDs or vascular aging (Figure 2).

## 4. mRNA Transcript Degradation: Role of Non-Coding RNAs (ncRNAs)

Early discoveries led to the concept that the three families of RNA jointly coordinate the process of protein synthesis are messenger (mRNA), transfer (tRNA), and ribosomal RNA (rRNA). However, it is well established that ~35% of the transcripts are coding RNA or mRNA that encodes for a protein, and the majority ~65% of transcripts are considered non-coding RNA or ncRNA [41]. The readers are directed to detailed reviews on ncRNA [41,42,43]. The ncRNA are further divided based on their size, location, and cellular function, and ncRNA of ~20–22 bases are called miRNA, and ncRNAs more than 200 bases in length are called lncRNA. The readers are directed to read detailed reviews on microRNA (miRNA) and long non-coding RNAs (lncRNAs) [42,43,44,45,46]. MiRNAs are essential regulators of gene expression. The dynamic interaction of miRNA with their target mRNAs is dependent on several factors, including affinity of the interaction, abundance of the target mRNA, and the intracellular location of miRNA and mRNA [45]. The biogenesis of miRNA can be canonical or non-canonical and is reviewed in detail previously [42,46]. Briefly, in the canonical pathway, the pri-miRNA transcript is cleaved by the microprocessor complex consisting of Drosha and DiGeorge Syndrome Critical Region 8 to become pre-miRNA [47]. After the pre-miRNA is exported to the cytoplasm and processed to be the mature miRNA, one of the strands is loaded to the Argonaute proteins (AGO) to form the miRNA-induced silencing complex (miRISC) to target mRNA of interest [48,49]. The non-canonical pathway starts with short hairpin RNA [50], which will be cleaved, exported, and processed with AGO2, but without DICER [51]. The miRNAs can bind to homologs 3′UTR (untranslated region), 5′UTR, or the coding region to inhibit the target gene expression and/or activity level [52,53]. MiRNA was first linked to autophagic regulation in 2009, when Zhu et al., reported that miR30 can post-transcriptionally regulate the expression of beclin1, an important regulator of autophagy [54]. 

It was found that miR30 can negatively regulate the activity of Beclin 1 by binding to its 3′UTR region, and following autophagy activation via nutrient deprivation or following rapamycin treatment, miR30 expression was inhibited [54]. However, miR30 overexpression was associated with inhibition of Beclin1-dependent autophagy [54]. Since then, many more miRNAs were discovered to regulate autophagy. In human cells, miR34 is found to regulate autophagy by directly targeting BCL-2 [55], which promotes survival by inhibiting autophagy via inhibiting beclin1 [56]. Another target of miR34 is SIRT1, where miR34 binds to the 3′UTR region of SIRT1 [57], and suppresses its expression, thereby inhibiting the expression of various ATGs [57]. Yang et al., reported that overexpressing miR34 in vitro led to inhibition of autophagy via downregulating ATG9 [58]. In C. elegans, loss of miR34 function resulted in extended lifespan by an autophagy-specific pathway, and this phenotype was reversed when autophagy genes (ATG9, ATG4, Beclin1) were silenced using RNAi [58]. In addition, it was found that in mice undergoing caloric restriction, which induces autophagy, the expression of miR34 was upregulated [59]. MiR17 and miR20 share the seed region AAGUGC that is highly conserved between species [60], as well as the target gene SQSTM1 (p62) [61]. MiR17 and miR20 regulate autophagy via regulating P62, which is critical to the process of autophagy and gets degraded in the autophagic process [62]. A recent study found that miR204 reduction induced autophagy in cardiomyocytes under hypoxia-reoxygenation. The miR204-mediated activation of autophagy was attributed to the potential of miR204 to target the essential autophagy gene MAP1LC3 in cardiomyocytes [63].

The family of miR196 is expressed from the intergenic region of HOX genes [64], and was recently found to be linked to Crohn’s disease, which is an inflammatory bowel disease. It was found that miR196 was overexpressed in patients with Crohn’s disease, and the potential target of this miRNA was identified to be the coding region of immunity-related GTPase family, M gene (IRGM) [65]. IRGM is involved in initiating the innate immune response in the presence of foreign pathogens via regulating autophagy [66], and miR196 was found to downregulate the protective variant of IRGM contributing to Crohn’s disease. Another miRNA related to Crohn’s disease is miR106B, which targets the autophagy gene ATG16L1 [67]. ATG16L1 is involved in the elongation of the autophagosomal membrane, and it also interacts with ATG5-ATG12 in the late stage of autophagy to promote the lipidation of LC3I [68]. Loss and gain of miR106B were shown to upregulate and downregulate autophagy, respectively, by targeting ATG16L1 in different cells [69]. Other miRNAs implicated in the regulation of autophagy are, miR376A which targets and blocks the expression of Beclin1 and ATG4C, and miR181A, which targets ATG5 [70]. We recently identified a novel inverse correlation between miR-378–3p and autophagy in vascular cells (unpublished data).

Besides the critical roles of miRNAs in regulating autophagy by regulating the expression of mRNAs, long non-coding RNAs (lncRNAs) are also important players in this process and are linked to the pathobiology of many cardiovascular diseases. The most well-studied mechanism is “miRNA sponging”, which is the process when lncRNAs indirectly modulate the stability and expression of mRNAs by sequestering miRNAs that target these mRNAs to prevent their interaction with each other [71]. The IncRNAs participating in this process are referred to as competing for endogenous RNAs (ceRNAs), and the final consequence of their sponging miRNAs is often the activation or de-repression of downstream molecular pathways [71]. In this review, we will mainly focus on ceRNAs that interfere with miRNAs involved in the regulation of autophagy, and subsequently, their involvement in the pathobiology of various CVDs. 

One of the first recognized ceRNAs is autophagy promoting factor (APF), which was found to activate autophagy in I/R induced MI by targeting the miR-188-3p/ATG7 axis. Wang et al., found that miR-188-3p normally represses induction of autophagy and MI by regulating the expression of ATG7 and APF functions to directly interact with miR-188-3p to prevent its binding with ATG7 leading to increased ATG7 expression and autophagy induction in cardiomyocytes. As a result, cardiac function was improved in mice with IR-induced MI treated with APF [72]. Yin et al., found that an IncRNA called GATA1 activated IncRNA can directly interact with miR-338 to prevent its interaction with the target ATG5 transcript leading to increased ATG5 expression and autophagy in murine cardiomyocytes under anoxia condition [73].

The ceRNA cardiac hypertrophy-related factor (CHRF) has pro-hypertrophic effects via targeting miR-489, a miRNA that normally suppresses the development of cardiac hypertrophy. Overexpressing miR-489 in cardiomyocytes resulted in reduced hypertrophic responses. CHRF antagonized the effects of miR-489 and promoted cardiac hypertrophy in vivo [74]. The lncRNA human large intergenic ncRNA ROR (lncRNA-ROR) was found to sequester miR-133 [75], which affects the MAPK signaling pathway that is closely linked to the transcription of cardiac genes [76]. Overexpression of lncRNA-ROR promoted cardiac hypertrophy in mice [75].

An IncRNA known to be involved in regulating autophagy via miRNA sponge and leading to CVDs is metastasis-associated lung adenocarcinoma transcript (MALAT1), the expression of which can be induced by hypoxia, high glucose, and oxidative stress. Fentanyl is known to have cardioprotective effects in I/R injury, and it was found MALAT1 sequesters miR-145 to prevent it from mediating cardioprotective effects of fentanyl as overexpressing MALAT1 or silencing miR-145 reversed the cardioprotective phenotype of cardiac muscle cells under hypoxia-reoxygenation condition [77]. MALAT1 is also involved in regulating the electrophysiology of cardiomyocytes and the pathobiology of arrythmias. A study found that MALAT1 “sponges” miR-200c to prevent it from interacting with high-mobility group box 1 (HMGB1), leading to reduced expression of subunits of potassium channels and disrupted cardiac electric current [78]. Besides sponging miR-200c, MALAT1 also binds to miR-26b, which normally targets ULK2, a protein involved in the early stage of autophagy. Li et al., found that MALAT1 has a protective role in brain microvascular endothelial cell (BMEC) injury caused by I/R by enhancing BMEC autophagy indicated by increased LC3 puncta, and downregulating the expression of miR-26b, which inhibits BMEC autophagy [79]. MALAT1 also has a protective role in MI. It was found that MALAT1 sponges miR-558, which normally targets ULK1 and suppresses isoproterenol-induced protective autophagy in MI, the result is increased autophagy and improved myocardial function in MI mice treated with MALAT1 [80]. On the other hand, MALAT1 facilitates the progression of coronary atherosclerosis by targeting the miR-15b-5p/MAPK1 axis leading to mTOR activation and autophagy induction. Treating mice with MALAT1 antagomir resulted in decreased autophagic influx and no atherosclerosis formed in mice [81].

A recent study by Liang et al., found that the IncRNA 2810403D21Rik/Mirf sequesters miR-26a, which normally targets ubiquitin-specific peptidase 15 (Usp15), to induce autophagy [82]. Inhibiting the expression of 2810403D21Rik/Mirf led to increased miR-26a and improved cardiac function in MI mice. In a rat model of diabetic cardiomyopathy (DCM), Feng et al., found that the IncRNA named DCM related factor (DCRF) acts as a miR-551b-5p sponge to induce cardiomyocyte autophagy. Inhibiting the expression of DCRF resulted in decreased autophagic activity, attenuated myocardial fibrosis, and improved cardiac function in DCM mice [83]. TGFB overlapping transcript 1 (TGFB2-OT1) was found to regulate autophagy in vascular ECs and is involved in vascular inflammation via sponging miR-3960, miR-4488, and miR-4459, which normally target ATG13. It was found that the expression of TGFB2-OT1 was induced by oxLDL, a common stimulus for vascular inflammation [84].

LncRNA can also directly regulate the transcription of autophagy-related genes and other genes involved in CVDs. For example, it was recently found that the lncRNA cardiac hypertrophy associated transcript (Chast) can negatively regulate the transcription of Pleckstrin homology domain-containing protein family M member 1 (Plekhm1), which regulates the fusion of autophagosome and lysosome during the autophagic process. Depletion of Plekhm1 led to blockage of lysosomal degradation of protein cargos [85]. Chast is activated by the nuclear factor of activated T cells (NFAT), which is a pro-hypertrophic transcription factor. Chast is transcribed from the opposite strand of the coding region of Plekhm1 and represses the transcription of Plekhm1. Overexpression of Chast resulted in decreased Plekhm1 and cardiac autophagy, leading to cardiac hypertrophy in mice, and silencing Chast resulted in attenuated TAC-induced cardiac remodeling [86]. Nexilinc (NEXN) is an F-actin binding protein involved in cell migration and cell adhesion and is believed to have an atheroprotective role [87]. Transcription of its antisense strand results in the lncRNA named NEXN antisense RNA 1 (NEXN-AS1) [87]. The microarray analysis showed that both NEXN and NEXN-AS1 expression were decreased in patients with atherosclerosis [87]. NEXN-AS1 interacts with the 5′ flanking region of NEXN and increases its transcription leading to downregulation of monocyte recruitment, secretion of adhesion molecules, and inflammatory cytokines by ECs. Silencing NEXN in ApeE (-/-) mice accelerated the process of atherosclerosis and the abundance of macrophage. Another lncRNA that potentially promotes atherosclerosis is the macrophage-expressed Liver X receptor-induced sequence (MeXis). Liver X receptor is a nuclear receptor activated by steroids and modulates transcription of genes important in cholesterol homeostasis. The expression of MeXis is increased upon activation of oxidized LDL and other lipid signals. It was found that MeXis regulates the LXR-dependent transcription of the Abca1, a gene involved in the regulation of cholesterol efflux. Silencing MeXis in mice resulted in impaired regulation of cholesterol and promoted atherosclerosis [88].

The lncRNA cardiac autophagy inhibitory factor (CAIF) has been recently linked to autophagy regulation and MI by indirectly regulating the transcription of myocardin. Myocardin is specifically expressed in cardiac and smooth muscle cells, and its expression is induced by p53. Myocardin is found to promote MI by inducing cardiac autophagy via targeting beclin 1. CAIF can downregulate the expression of myocardin by binding to p53, which prevents their interaction with each other. Decreased myocardin leads to decreased autophagic activity and is found to attenuate I/R induced cardiac injury in mice [89]. Zhuo et al., found that in a rat model of diabetic cardiomyopathy, expression of the lncRNA H19 was decreased. The subsequent study done by the same authors found that overexpressing H19 resulted in decreased cardiac autophagy by indirectly regulating transcription of DIRAS3, a GTP binding protein belonging to the RAS family. H19 interacts with EZH2 to reduce H3K27me3 binding in the promoter of DIRAS3, which increases phosphorylation of mTOR and subsequent autophagy inhibition [90]. The role of miRNA and lncRNAs on autophagy regulation is important, and our understanding of their relationship is a critical step toward understanding the epigenetic regulation of autophagy by ncRNAs.

## 5. mRNA Transcript Degradation: Role of Regulated mRNA Decapping/Degradation

The 5′end cap on eukaryotic mRNA is required for maintaining mRNA stability, increasing the accuracy of pre-mRNA splicing, and to prevent its degradation by endonucleases [91]. In the cytoplasm, 5′cap is recognized by the translation initiation factor eIF4E, which is a critical step for consequent mRNA translation. Removal of the cap is an important regulatory event that controls transcript expression. The decapping process is catalyzed by a series of specific decapping enzymes to modulate different subsets of mRNAs expression, and mRNA without a cap is unstable and degraded by exonucleases (Figure 3). Decapping mRNA 2 (Dcp2) is the first discovered decapping enzyme that controls the majority of decapping events in eukaryotes. Hu et al., found that in both yeast and mammalian cells, Dcp2 gets phosphorylated via TOR-dependent pathway and associates with RNA helicase RCK family to form a complex with ATG mRNAs to promote their decapping and subsequent autophagy inhibition [92]. Vad1 is another decapping enzyme found in mammals, and suppresses autophagy via decapping of gene ATG8 [93]. Lucas et al., studied 14 patients with mutations in the PIK3CD gene, who displayed hyperactive mTOR activity leading to an increased decapping and degradation of ATG transcripts causing autophagy reduction [94]. All these studies suggested a close link between decapping of ATG transcripts and regulation of autophagy, warranting further research. 

## 6. mRNA Transcription: Role of Histone Modifications

Chromatin is a complex of condensed DNA scaffold closely wounding around histones and other associated proteins. The structure of chromatin is regulated by the post-translational modification of histones, which are a class of alkaline proteins that support the packaging of DNA into nucleosomes [95]. Two major histone modifications that are linked to the regulation of autophagy are acetylation and methylation (Figure 3). Histone acetylase (HAT) transfers an acetyl group to the epsilon amino acid on the positively charged lysine residues on histone, which weakens the interaction between histone and DNA, making DNA more accessible for transcription and translation [96]. Histone deacetylases (HDAC) increase the electrostatic charges of histone and renders DNA more condensed [97]. Sirtuin (SIRT) is an NAD+ dependent HDAC, and SIRT1 is a well-studied modulator of autophagy that has long-term beneficial effects on aging [98]. SIRT1 activation is essential for induction of autophagy, and mechanistically, under cellular stress, such as nutrient deprivation, SIRT1 activity increases, and it directly interacts with and deacetylate autophagy-related proteins (ATG5, ATG7, LC3) to induce autophagy [99]. The regulation of autophagy by SIRT1 can be understood by the fact that SIRT1-deficient mice and SIRT1-deficient cells displayed significantly reduced autophagic activity in vivo and in vitro [99], respectively. Furthermore, SIRT1 overexpression rescued the autophagy state in those mutant cells [99]. SIRT1 can also regulate autophagy by deacetylating forkhead box O-3 (FoxO3), a transcription factor for various autophagy-related genes that are critical for autophagy induction [100]. The primary deacetylation substrate of SIRT1 is lysine 16 on histone 4 (H4K16) [101], which suppresses transcription of autophagy-related genes, including ULK1, ATG9, ATG8, and VMP1, leading to decreased autophagic influx [7]. Research has confirmed the histone acetylatransferase HMOF/KAT8 that can antagonize the activity of SIRT1, promoting DNA decondensation and promotes autophagy induction [7]. Hajji et al., has found that overexpressing SIRT1 can overcome the autophagy-inducing effect of HMOF/KAT8 and reduce the basal level of autophagy [102]. Autophagy is reported to be a double-edged sword, where it can promote survival or apoptosis, in a context-dependent manner [103]. Accordingly, overstimulation of autophagy can lead to cell death which can potentially be prevented by downregulation of H4K16 acetylation that suppresses expression of ATGs [7]. Moreover, SIRT1 can also physically interact with mTOR inhibitor TSC2 to regulate autophagy [104]. Overall, it is now well established that SIRT1 is a critical molecular switch for epigenetic regulation of autophagy. The effect of HDAC is cell type-specific, since inhibiting HDAC results in autophagy induction in yeast, glioblastoma, and MEFs, while it suppresses autophagy in cardiomyocytes [105].

Besides acetylation, methylation of histone, especially H3, is also important for autophagy regulation. For example, the methytransferase EHMT2/G9A combines with enhancer of zeste homolog 2 (EZH2), which belongs to the family of polycomb-repressive complex (PRC), to target and methylate H3K9 resulting in inhibited basal autophagy by activating the expression of mTOR [106]. Upon autophagy induction, EHMT2/G9A dissociates from the promoter that is responsible for regulating transcription of essential autophagy genes, and this results in decreased methylation and thereby increased activation of H3K9 that promotes transcription of autophagy genes [107]. It was shown that in human cell lines, when there is increased activation of H3K9, there is an increase in expression of various important genes involved in autophagy (ATG18, ATG9, BNIP3, ATG8/LC3B) [107]. While EHMT2/G9A was found to methylate H3K9, KDM2B and KDM1A demethylase remove the methyl group from H3K9 and lead to activation of autophagy via inhibiting mTOR [101]. Another H3 modification associated with autophagy is the methylation of histone 3 arginine 17 (H3R17) by co-activator-associated arginine methyltransferase 1 (CARM1). Under starvation, there is increased AMPK-dependent phosphorylation of FoxO3, which leads to increased CARM1 allowing transcription of autophagy-related genes [108]. Modifications on histone 4 (H4) are also shown to regulate autophagy. For example, Nutrient deprivation is associated with an increase in histone 4 lysine 16 (H4K16) acetylation and histone 4 lysine 20 (H4K20) methylation leading to increased and decreased RNA polymerase II activity, respectively [109], affecting the expression of ATGs and thereby autophagy [110].

lncRNAs can modulate epigenetic markers on important genes involved in autophagy and CVDs. Hu et al., found that in cardiomyocytes isolated from mice with hypoxia/reoxygenation induced injury, the lncRNA MALAT1 represses the transcription of TSC2 by recruiting EZH2 to increase H3K27me3 modification in the TSC2 promoter region. Inhibiting TSC2 expression activates mTOR signaling and induces autophagy. They also found that MALAT1 expression enhanced apoptosis by inhibiting autophagy via regulating the TSC2-mTOR signaling pathway [111]. MALAT1 is also shown to interact with HDAC9 and the chromatin-remodeling enzyme BRG1 in VSMCs to regulate transcription of contractile genes involved in aortic aneurysm. MALAT1 functions by recruiting PRC2 and increasing methylation of histone H3 in the promoter regions of those genes leading to decreased transcription, and decreased contraction [112].

## 7. Epigenetic Regulation in CVDs

NcRNAs are essential regulators of various biological processes and have been increasingly linked to CVDs, including heart failure (HF), hypertrophy, arrythmias, myocardial infarction (MI), contractility defects, and inherited cardiomyopathies. Thus, ncRNAs represent promising targets for the treatment of CVDs. In addition, miRNAs have been found in blood circulation and investigated for their potential as novel biomarkers that could be used in combination with existing prediction models to increase the predictive accuracy of disease prognosis.

### 7.1. Atherosclerosis

Atherosclerosis is a condition when plaques build up and accumulate in arteries [113]. It is commonly triggered by hypercholesterolaemia or an increase in the level of cholesterol in plasma, which leads to a change in endothelial cell membrane permeability, allowing LDL particles to enter the arterial wall [113]. Monocytes are recruited to the subendothelial space and turned into foamy macrophages, which further promotes the accumulation of cholesterol, and finally, differentiation of vascular smooth muscle cells (VSMCs) [114]. Endothelial dysfunction is a major contributing factor to the initiation of atherosclerosis, since it was found that the disturbed laminar flow caused by plaque formation compromises endothelial function. NcRNAs are shown to regulate atherosclerotic progression; such as miR-126-5p prevents the formation of atherosclerotic plaque by maintaining a proliferative reserve of ECs by suppressing the Notch1 inhibitor delta-like 1 homolog [115]. Another important step during atherosclerosis is the migration of VSMCs from the media to the intima. MiR145 is highly expressed in VSMC and regulates its plasticity and differentiation [116]. We previously demonstrated that that miR145-overexpression in ApoE^-/-^ mice reduced plaque size and increased stability [117]. The IncRNA H19 has been found to disrupt the endothelial cell, which ultimately leads to atherosclerosis. Guo et al., found that the lncRNA FA2H-2 is associated with atherosclerosis by binding to the promoter of mixed lineage kinase domain-like protein (MLKL) gene, which leads to autophagy inhibition via the mTOR signaling pathway. Furthermore, loss of FA2H-2 or overexpressing MLKL led to an exacerbation of oxidized LDL-induced inflammatory responses [118].

### 7.2. Cardiac Contractility Defects

Cardiac contractility provides the heart with the mechanical force to beat, and defects in contractility cause heart diseases. Myosin heavy chains (MHCs) are the major molecular motor force in muscle cells [119], and there are two MHC isoforms specifically expressed in the mammalian myocardium [120]. MHCα is encoded by the gene myh6, and MHCβ is encoded by the gene myh7. Depending on the type of isoform expressed at a certain time, heart contractility, mechanical performance, and energy turnover vary [120]. MHCβ expression is higher in the developing embryonic heart [119], and higher expression of myh7 in the adult heart is one of the major causes of cardiac hypertrophy. Hang et al., found that Brg1, which is a chromatin-remodeling protein interacts with HDAC and poly ADP ribose polymerase (PARP) to form a complex that maintains embryonic cardiomyocytes in a fetal differentiation state by repressing MHCα and activating MHCβ [121]. Transcription of the antisense strand of myh7 results in a lncRNA. A recent study done by Han et al., showed that IncRNA779, which they named myosin heavy chain associated RNA transcript 779 (Mhrt779), can bind to the helicase domain of Brg1 and prevents it from binding to the myh7 promoter under pre-stressed condition [122]. When cardiomyocytes are stressed, Brg1 expression increases, forms a complex with HDAC/PARP, and outruns the inhibitory effect of Mhrt779 so that it can inhibit myh6 and promote expression of myh7 promoting hypertrophy; and inhibition of Mhrt779 prevented experimental cardiac hypertrophy in mice [122]. Recently, miRNAs were also found to regulate myh6 and myh7 expression [123]. It was shown that in mouse embryonic stem cells and neonatal rat ventricular cardiomyocytes, miR-27a stimulated expression of myh7 upon initiation of thyroid hormone receptor signaling and has no effect on myh6 expression. The role of miR-27a in regulating myh7 expression was further supported in the in vivo study using experimental hypertrophic hearts in mice, where a simultaneous upregulation of miR-27a and myh7 expression was observed [123]. MiR-208a that is encoded by the intronic space of myh6, was also found to regulate MHC gene expression [124]. Under cardiac stress, loss of miR-208a resulted in downregulation of myh7 expression [124]. Moreover, following upregulation of miR-208a in type-II diabetic mice hearts, there was an increase in expression of MHCβ, which is a precursor of cardiac hypertrophy and inhibiting the expression of miR-208a resulted in decreased MHCβ activation and development of cardiac hypertrophy [125]. Another mechanism important for heart contractility involves sarcoplasmic reticulum Ca^2+^ ATPase activity (Serca2a) [126]. It was found that in mice hearts under cardiac stress, there was a reduction of Serca2a expression caused by a reduction in expression of miR-22 [126]. Clinically, reduced Serca2a marker is commonly associated with HF [127]. A recent study found that miR-25 also targets and inhibits the expression of Serca2a, and inhibition of miR-25 using an antagomir successfully restored the cardiac function in a mouse model of HF [128]. Furthermore, miR-1 was found to be associated with contractility defects via downregulating genes involved in sarcomere assembly, including cardiac calmodulin and myosin light chain kinase, leading to sarcomeric disassembly and disturbed heart function [129]. Overexpressing miR-1 in rat ventricular cardiomyocytes led to abnormal Ca^2+^ influx and decreased expression of protein phosphatase 2A, which is a critical factor that affects cardiac contractility leading to cardiac arrythmias [130]. Another phosphatase involved in the pathology of arrythmias is phosphatase and tension homolog (Pten), which is the target of the miR-17-92 cluster. Overexpressing this cluster in a transgenic mouse model led to spontaneous arrythmias [131]. Furthermore, it was suggested that overexpressing miR-133 and miR-1 expressions could prevent the development of cardiac hypertrophy, and inhibiting miR-133 expression induced cardiac hypertrophy in mice [132].

### 7.3. Heart Failure

Heart failure occurs when there is not enough circulatory forces generated by the heart to support the metabolic needs of the body [133]. The circulating miRNAs have been identified in plasma from patients with HF. Particularly, miR-1 and miR-21 expressions were downregulated and upregulated, respectively, in patients with HF [134]. In addition, miR-423 [135], and miR-126 expressions were downregulated in HF patients, and miR-1254 and miR-1306 expressions are positively associated with risk of death and hospitalization in acute HF patients [136].

#### 7.3.1. Myocardial Infarction

Myocardial infarction is characterized by decreased blood flow to the heart leading to hypoxia in cardiomyocytes, while acute MI caused by ischemic injury and cardiomyocyte death can lead to structural alterations and cardiac dysfunction [137]. I/R injury can be induced during the process of revascularization leading to more cardiac injury [138]. MiRNA tightly regulates the proliferation and regeneration of cardiomyocytes following MI. MiR-133 and miR-1 are the most abundant miRNAs in cardiac tissue responsible for myoblast proliferation and differentiation [132]. The circulatory miR-1 level was found to peak 6 h post-infarction and returned to normal levels three days after, and patients with higher serum miR-1 level had larger infarct size [139]. MiR-133 is also important for cardiac development, but its relationship with MI is controversial [140,141,142,143]. On the one hand, miR-133-overexpressing mesenchymal stem cell transplantation in experimental-MI showed improved cardiac function, smaller infarct size, and lower inflammatory responses compared to controls. It was later found that miR-133 exerted its protective role via repressing the expression of snail 1 or fibrinogenesis-promoting gene 1 [140]. Another study found that the beta-blocker carvedilol reduced oxidative stress-induced cardiomyocyte apoptosis in infarcted hearts via upregulating miR-133 expression [141]. A more recent study done by Yu et al., found that Aloe-emodin (AE), which is an anthraquinone, improved cardiac function and reduced infarct size in mice via upregulating miR-133 and suppressing caspase-3 activity [142]. However, studies in humans did not find any significant differences in the serum level of miR-133 in patients with MI and healthy controls [143]. Other miRNAs that were found to be associated with MI are miR-21 [144]. MiR-208a (Biatek) and miR-499 [144] as significantly higher plasma levels of these miRNAs were detected in MI patients compared to healthy controls [144,145].

#### 7.3.2. Cardiac Hypertrophy and Cardiac Fibrosis

Cardiac hypertrophy refers to the condition when the heart muscles are abnormally enlarged [109]. The IncRNA terminal differentiation-induced ncRNA (TINCR) interacts with EZH2, a methyreansferase that regulate expression of calmodulin-dependent protein kinase II sigma (CamKII). Forced expression of CamKII induces cardiac hypertrophy and HF in mice [109]. Overexpression of TINCR resulted in decreased expression of CamKII and attenuated angiotensin-II induced cardiac hypertrophy. Cardiac inflammation is one of the major factors that lead to cardiac fibrosis [146]. Septic myocarditis is caused by acute inflammation in the heart in response to invading pathogens and leads to left ventricular systolic dysfunction [147]. LncRNA named HOTAIR is associated with septic myocarditis by promoting the expression of NFkB and tumor necrosis factor alpha, which are pro-inflammatory cytokines released during a septic shock. Thus, silencing HOTAIR might be a potential therapeutic treatment for septic myocarditis [148]. The lncRNA maternally expressed gene 3 (MEG3), is implicated in cardiac fibrosis. Following pressure overload, cardiac fibroblasts (CFs) respond by driving extracellular matrix remodeling, which leads to cardiac fibrosis. It was found that MEG3 was downregulated during late cardiac remodeling, and silencing MEG3 resulted in decreased expression of metalloprotease-2 (MMP-2). MMP-2 plays critical roles during cardiac remodeling by regulating the activity of p53. Inhibition of *Meg3* in vivo decreased MMP-2 expression and p53 activation and attenuated cardiac fibrosis [149]. MEG3 was also found to be associated with autophagy [150]. Mycobacterial infection, causes autophagy induction and Meg3 downregulation in macrophages [150], however the relationship between Meg3, autophagy, and cardiovascular diseases needs to be further explored. The lncRNA nuclear-enriched abundant transcript 1 (Neat1) was found to be highly expressed in ischemia reperfusion-treated diabetic rat myocardial tissues, and high Neat1 expression exacerbated myocardial I/R injury by inducing autophagy via upregulating FoxO1 [151].

## 8. Therapeutic Potential of ncRNA in CVDs

There has been immense progress in the areas of diagnostic, as well as the treatment of cardiovascular diseases, but it remains the number one cause of death worldwide, warranting further improvements [12]. The significant involvement of miRNA and lncRNA in the pathobiology of CVDs suggests their potential for diagnostic and therapeutic use. Li et al., reported that overexpressing miR-99a using a mimic improved cardiac function in a MI murine model by inducing autophagy [152]. In addition, miR-99a overexpression also decreased apoptosis induced by hypoxia and improved cardiac function in the ischemic heart in mice [152]. Studies have found that the miR-212/132 family specifically targets the FoxO3 transcription factor, which normally inhibits hypertrophy and promotes autophagy [153]. Overexpressing these miRNAs lead to impaired autophagy and inhibiting the expression of miR-212/132 via injection of antagomirs rescued cardiac hypertrophy and HF in mice [153]. Another study found that pharmacological inhibition of miR-652 expression in the cardiac hypertrophy mouse model improved cardiac function and attenuation of cardiac hypertrophy [154]. In an adult porcine model of percutaneous I/R, local delivery of locked nucleic acid-modified antisense miR-92a resulted in a reduced infarct size and improved left ventricular end-diastolic pressure [155]. Rayner et al., found that inhibiting miR-33A and miR-33B in non-human primates resulted in reduction of very low-density lipoprotein (VLDL) and elevation of high-density lipoprotein (HDL), revealing the potential of miR-33A/B in treating atherosclerosis [156]. NcRNAs are also shown to demonstrate sex-related differences. For example, miR-34a inhibition in cardiomyopathy was found to be more effective in females than in males [154]. It is important to have a better understanding of the interaction between miRNA and the complex biological system to improve the therapeutic efficiency, as well as to prevent potential side effects. Recently, circulating miRNAs have been associated with various CVDs and recognized for their potential as novel biomarkers to assist with clinical prediction models. We conducted a genome-wide circulatory miRNA sequencing in patients with peripheral artery disease (PAD), and found that miRNA-1827 expression level directly correlated with the PAD severity in patients, which could serve as a potential biomarker for PAD [157]. Other potential prognostic markers have been identified in HF (miR-16, 27a, 101, 150, 122, 210, 499), and coronary heart disease (miR-126, 197, 233) [158]. However, there have been challenges associated with measuring the exact level of ncRNA to be utilized as a biomarker, due to the lack of standard protocols and references, which warrants further studies to improve the practical utility of ncRNA biomarkers [159].

Although increasing attention has been put on exploring the therapeutic potential of miRNA to treat CVDs, not enough research has been done on other ncRNA types, such as lncRNA. LncRNA has been closely associated with pathways, such as autophagy regulation, proliferation, apoptosis [160]—all of which have implications in CVDs, indicate the plausible role of lncRNAs in CVDs. In order to establish an association between lncRNAs and CVDs, we have previously demonstrated the differential regulation of several lncRNAs in ECs following treatment with atherosclerosis-associated oxLDL and pravastatin [161]; diabetes-associated high glucose [162]; inflammation-associated lipopolysaccharides; and fibrosis-associated TGFβ [163]. These in vitro findings instigate lncRNAs with CVDs, however, more detailed studies are needed to elucidate the relationship of lncRNA, as well as its use as a biomarker or as a therapeutic target in CVDs. The main reason behind limited research on lncRNAs in CVDs are unlike miRNAs, lncRNAs are poorly conserved between species to obtain convincing in vivo result, one needs to find representative animal models. In addition, some of the lncRNAs are ubiquitously expressed in humans; and very few lncRNAs are characterized.

## 9. Future Perspectives

As learned from this review, impaired epigenetic regulation of autophagy can lead to CVDs, including heart failure, arrythmias, myocardial infarcts, contractility defects, and cardiac hypertrophy (Figure 1). For the regulatory role of ncRNAs, when they act on the transcription level, they can tightly control the expression of many autophagy-related genes leading to regulation of autophagy. NcRNAs can also influence histone modifications leading to either inhibition or stimulation of autophagy-related genes. The interconnection and regulation between ncRNA and autophagy, as well as their involvement in the pathobiology of various diseases, form an intricate biological network with the understanding that it could contribute to the development of more effective therapeutics. The therapeutic potential of ncRNAs has been explored by either overexpressing or inhibiting their expression and has shown promising results in restoring cardiovascular function and increasing survival rates in many in vitro studies and in animal models. From a clinical perspective, ncRNAs are also potential biomarkers and prognostic markers that could be used in combination with existing prediction models to increase predictive accuracy. However, our knowledge is still limited on the exact mechanisms of ncRNA regulation on autophagy, and there is also a lack of human clinical trials for testing the therapeutic efficacy of ncRNA-based therapy. In addition, the autophagic pathway and different modulators should also be explored in depth. More research needs to be done to obtain a better understanding of this complex biological interaction to better predict the side effects of these therapies, as well as to increase their efficacy. Larger cohorts of patients are needed to study ncRNA biomarkers to confirm the specificity of the disease. Although more research needs to be done before ncRNAs for treating CVDs can become a clinical reality, the promising results, as shown in cell and animal models, shine a light on their potential to open a new era for CVD treatment in the near future.

## Figures and Tables

**Figure 1 ijms-22-06544-f001:**
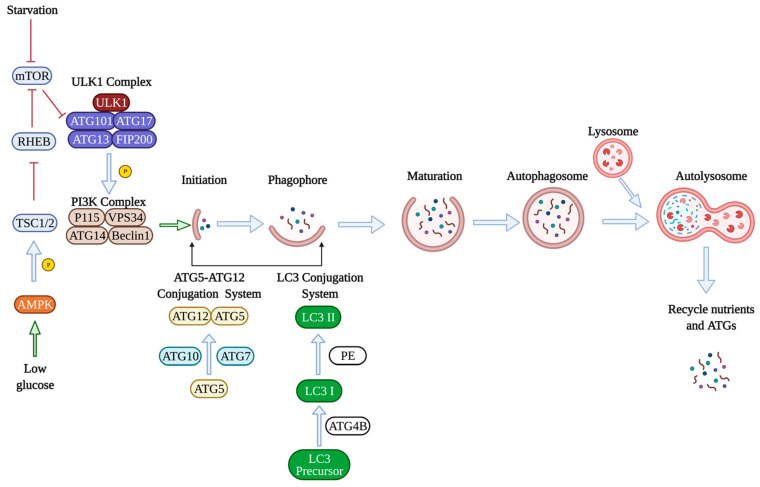
Molecular mechanisms of autophagy. Under nutrient-rich conditions, the mammalian target of rapamycin (mTOR) inhibits autophagy by inhibiting the unc-51-like kinase 1 (ULK1). The ULK1 complex consists of autophagy-related genes ATG101, ATG17, ATG13, and FIP200, which is a focal adhesion kinase bound to ULK1. ULK1 then phosphorylates other proteins, such as the class III phosphatidykinositol-3-kinase (Ptdlns3K) complex, which consists of vacuolar protein sorting 34 (VPS34), ATG14, Beclin1, and P115 protein. Adenosine monophosphate-activated protein kinase (AMPK) is another protein involved in autophagy induction. Under glucose deprivation, AMPK phosphorylates the tuberous sclerosis complex (TSC1/2), inactivates RHEB, which is a GTPase activating protein, and inhibits mTOR leading to autophagy induction. The formation and elongation of the autophagosomes involve two ubiquitin-like conjugation systems. The first conjugation system is the ATG5-ATG12 conjugation system formed by the protein ATG5 conjugating to ATG12 with the ubiquitin-like enzymes ATG7 and ATG10, which promotes the elongation of the autophagosomal membrane. The second conjugation system involves microtubule-associated protein 1 light chain 3 (LC3). LC3 is cleaved by the protease ATG4B into its mature form LC3-I, conjugating with phosphatidylethanolamine (PE) to form the lipidated version LC3-II. Finally, autophagosomes fuse with the lysosome to degrade the cargo by the lysosomal enzymes. Nutrients and ATGs are recycled for future use. Green arrows mean activate/promote, blue arrows indicate normal physiological processes, red means inhibition.

**Figure 2 ijms-22-06544-f002:**
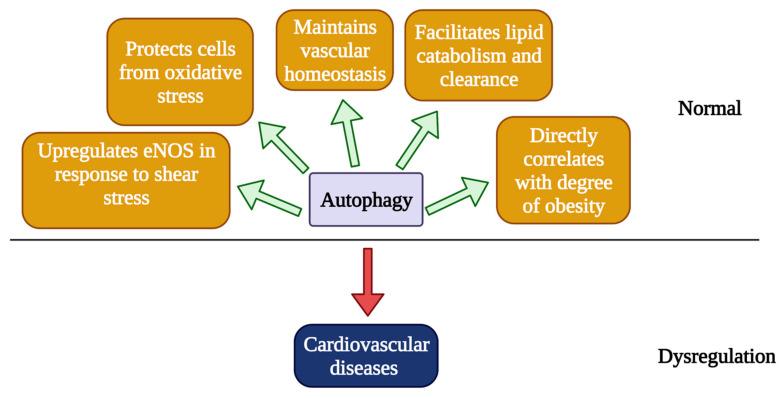
Roles of autophagy in the pathobiology of cardiovascular diseases. Autophagy is essential for regulating endothelial functions to maintain vascular homeostasis. Autophagy plays an important role in lipid clearance, and impaired autophagy leads to the accumulation of lipid, which is a known risk factor for the development of atherosclerosis. Mitophagy can protect vascular ECs from oxidative stress by removing damaged mitochondria and by preventing the elevation of ROS. Shear stress in the vessel wall can induce autophagy, which resulted in reduced ROS and increased NO production, mainly by upregulating eNOS and increased cell viability. Autophagy correlates with the level of obesity, a major risk factor for developing CVDs. Dysregulated autophagy can lead to CVDs, which will be discussed in detail in the next section.

**Figure 3 ijms-22-06544-f003:**
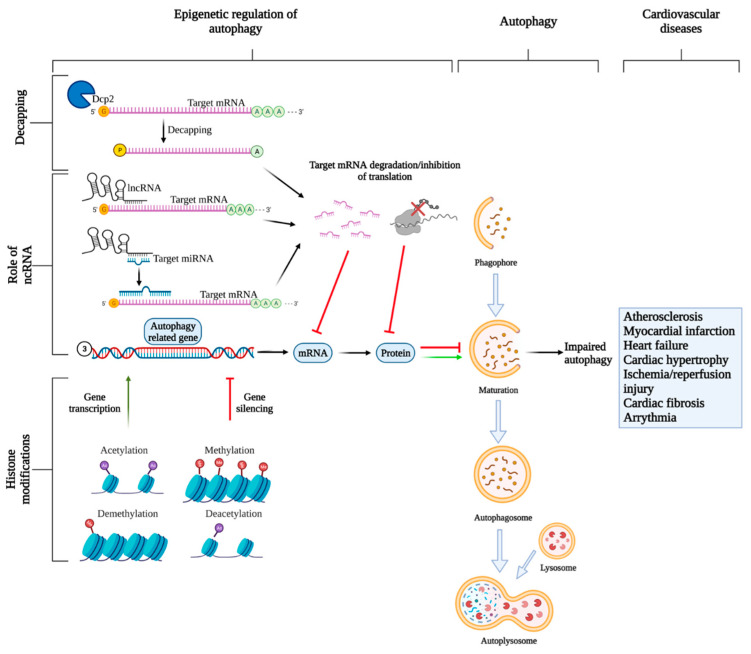
Epigenetic regulation of autophagy. Three types of epigenetic regulations are discussed in this review: mRNA decapping, regulation of gene transcription and translation by ncRNAs, and histone modifications. All three processes tightly control autophagy by regulating the expression of autophagy-related genes. The decapping enzyme Dcp1 is responsible for removing the 5′cap on the target mRNAs which leads to decreased mRNA stability and consequent degradation by exonucleases. NcRNAs, such as miRNAs, can bind to target mRNAs leading to their inhibition or degradation or inhibition depending on the degree of complementarity between miRNA and mRNA. Another ncRNA, lncRNAs can control autophagy-related gene expressions either directly by interacting with target mRNAs leading to their degradation, or indirectly by acting as “miRNA” sponges to sequester miRNAs to prevent interaction with their target mRNAs allowing gene expression. Lastly, histone modifications can also influence gene expressions. Acetylation and demethylation, in most cases, lead to gene transcription, while methylation and deacetylation mostly lead to gene silencing. Reduced expression of autophagy-related genes can lead to dysregulated autophagy, which is linked to CVDs.

## Data Availability

Not applicable.

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
