# Peer review of "Epigenetic Regulation of Autophagy in Cardiovascular Pathobiology"

_ijms, 2021, doi:10.3390/ijms22126544_

Round 1
Reviewer 1 Report
In this article, authors Bu et al have summarized and discussed the epigenetic regulation of biological process autophagy in cardiovascular diseases. Authors have elegantly explained each of the sections in a didactic manner, which is more useful for the readers; however, an addition of schematic representation for each section will make the readers continue reading the article. It is a well known fact that a review article with out a schematic cartoon will not bring enthusiasm to the readers. Authors are required to add the same to each section. Also, one schematic cartoon is required to demonstrate the pathways or molecular signals involved in the process of autophagy. Mitophagy is a process of autophagy. Authors could possibly expand the review to include that too. Authors have discussed the topic Autophagy under one umbrella - CVD. This is very vague. Authors should rewrite the sections to segregate the article in terms of diseases - For instance, Myocardial Infarction, Atherosclerosis, Cardiomyopathy and etc. Readers will also look for a disease specific discussion. I understand, it is a whole lot of rewrite. But, that is recommended and required to facelift this review article.
Author Response
Dear reviewer, thanks so much for your insightful comments! We have made changes to the review accordingly. Two new figures are now added, one for the detailed autophagic mechanisms and other for the section about autophagy in cardiovascular pathobiology. New information on mitophagy is now added. Sections on epigenetic regulation have been segregated into specific diseases. Thank you!
Reviewer 2 Report
Dear Authors,
I found this reviewed article entitled Epigenetic Regulation of Autophagy in Cardiovascular Pathobiology interesting.
Starting to work out the epigenetic regulation of autophagy proves an excellent knowledge of the issues and current global scientific trends. I read the article with attention and interest. I believe that all the issues relevant to the presented issue have been taken into account. An interesting addition is a figure, which makes it easier for a reader unfamiliar with the subject to understand the essence of the issues raised. I believe that the article does not need to be extended with new issues; it is comprehensive, properly prepared and does not require changes on the part of the authors. In the present form, they recommend the article for consideration for publication in IJMS.
Best regards
Author Response
Dear reviewer, we are more than thrilled that you enjoyed reading this review and thank you for your thoughtful comments. We have now added two new figures to the review, one for the detailed autophagic mechanisms and one for the section about autophagy in cardiovascular pathobiology. Thank you!
Round 2
Reviewer 1 Report
No more suggestions